# Nosocomial Pneumonia Caused in an Immunocompetent Patient by the Emergent Monophasic ST34 Variant of *Salmonella enterica* Serovar Typhimurium: Treatment-Associated Selection of Fluoroquinolone and Piperacillin/Tazobactam Resistance

**DOI:** 10.3390/antibiotics11030303

**Published:** 2022-02-24

**Authors:** Xenia Vázquez, Lorena Forcelledo, Salvador Balboa-Palomino, Javier Fernández, María Rosario Rodicio

**Affiliations:** 1Área de Microbiología, Departamento de Biología Funcional, Universidad de Oviedo (UO), 33006 Oviedo, Spain; xenia_grao@hotmail.com; 2Instituto de Investigación Sanitaria del Principado de Asturias (ISPA), 33011 Oviedo, Spain; lforcelledoespina@yahoo.es (L.F.); javifdom@gmail.com (J.F.); 3Servicio de Medicina Intensiva, Hospital Universitario Central de Asturias (HUCA), 33011 Oviedo, Spain; salvabalboa91@gmail.com; 4Servicio de Microbiología, Hospital Universitario Central de Asturias (HUCA), 33011 Oviedo, Spain; 5Research & Innovation, Artificial Intelligence and Statistical Department, Pragmatech AI Solutions, 33003 Oviedo, Spain; 6Centro de Investigación Biomédica en Red-Enfermedades Respiratorias, 28029 Madrid, Spain

**Keywords:** nosocomial pneumonia, gastrointestinal infection, *Salmonella enterica*, ST34, fluoroquinolone resistance, piperacillin/tazobactam resistance

## Abstract

The present report describes an uncommon case of nosocomial pneumonia caused by *Salmonella*
*enterica* in an immunocompetent patient. The patient was admitted to ICU of a tertiary hospital due to low level of consciousness, aphasia and seizure episodes. Four days after hospitalization, he developed nosocomial pneumonia, which evolved into septic shock. Gram-negative bacilli were recovered from blood, tracheal aspirate and fecal samples of the patient. The isolates, which were identified as *Salmonella enterica*, proved to be resistant to ciprofloxacin, amoxicillin/clavulanic acid and piperacillin/tazobactam. Four months before, the same bacterial species was recovered from feces and blood cultures of the patient, admitted to the nephrology ward of the same hospital with diagnosis of gastroenteritis and acute renal failure. However, at that time, the isolates were susceptible to the above-mentioned antibiotics. Genome sequencing revealed that all isolates were closely related and belonged to the emergent ST34 monophasic variant of *S. enterica* serovar Typhimurium. Since the patient has received therapy with fluoroquinolones and amoxicillin/clavulanic acid, these results support treatment-associated selection of the acquired resistances. In conclusion, this case represents a paradigm of selective pressure leading to in vivo development of resistance to highly relevant antibiotics, including the piperacillin/tazobactam combination used for empirical management of severe infections at ICU.

## 1. Introduction

Non-typhoidal serovars of *Salmonella enterica* (NTS) are among the most common foodborne pathogens, posing a considerable threat to public health, which is worsened by the emergence of antimicrobial drug resistance [1]. In humans, NTS usually cause self-limiting gastroenteritis, characterized by diarrhea, abdominal cramps, nausea, vomiting and fever. With a low frequency, these bacteria can spread beyond the intestine, giving rise to extra-intestinal focal and invasive infections, which mostly occur in children, the elderly or immunocompromised patients [2]. Third-generation cephalosporins and fluoroquinolones are drugs of choice for the management of serious *S. enterica* infections in adults [3]. Here, we report a rare case of nosocomial pneumonia caused by *S. enterica* in an immunocompetent patient admitted to the intensive care unit (ICU) of a tertiary hospital, with concomitant in vivo development of fluoroquinolone and piperacillin/tazobactam resistance.

## 2. Case Presentation

A 71-year-old male was admitted to ICU of a tertiary hospital in Northern Spain, due to low level of consciousness, aphasia and seizure episodes. The patient had a medical history of hypertension, diabetes, a transient ischemic attack and mild chronic renal failure, with no immunosuppressive medication or disease. A computerized tomography scan was performed showing a subdural hematoma, subarachnoid hemorrhage and other hematomas in frontal and occipital lobes with a traumatic origin, which also resulted in two right ribs being broken. It was unknown whether the trauma occurred before or during a seizure episode. At admission, the patient presented a slight recovery staying with Glasgow coma scale 13 and no other symptoms. Four days later, he developed respiratory failure and fever with leukocytosis, high procalcitonin levels (46 ng/dL; normal range 0.0–0.5 ng/dL) and new infiltrates in the right inferior lung lobe, which appeared in the chest radiography. The patient was intubated and diagnosed of early nosocomial pneumonia according to international guidelines [4], and antimicrobial therapy with amoxicillin/clavulanic acid was started. Gram-negative bacilli were recovered from a blood culture incubated in a BacT/ALERT^®^ (bioMérieux, Marcy l’Etoile, France) detection system, and also from a tracheal aspirate where the Gram stain revealed a large number of leukocytes, a small number of epithelial cells and a single type of Gram-negative bacilli. Twenty-four hours later, parameters related to the number of leukocytes, C-reactive protein and to creatinine and urea levels worsened, and liver function abnormalities with cholestasis were observed in blood biochemical tests. The patient evolved to septic shock and antibiotic treatment was escalated to meropenem. Abdominal scan was performed with no pathological findings apart from cholelithiasis. 

About 10^9^ colony-forming units (CFU)/mL of the Gram-negative bacilli detected by Gram stain grew in agar McConkey (BioMérieux, Marcy l’Etoile, France) from the respiratory sample, and they were identified as *S. enterica* by matrix-assisted laser desorption/ionization time-of-flight mass spectrometry (MALDI-TOF MS; Bruker Daltonics, BD, Bremen, Germany). The same bacterium was recovered from the blood culture after subculture in agar TSA (bioMérieux). Antimicrobial susceptibility testing, accomplished by the Microscan System (Beckman Coulter, Brea, CA, USA) and interpreted according to EUCAST guidelines (www.eucast.es, accessed on 30 June 2020), showed that the isolates were susceptible to all antibiotics tested except amoxicillin/clavulanic acid, piperacillin/tazobactam and ciprofloxacin (also, the pefloxacin disk diffusion assay recommended by EUCAST to screen for clinical fluoroquinolone resistance was positive) (Table 1). According to this, the antibiotic therapy was de-escalated to cefotaxime. Even though there was no diarrhea, a stool sample was taken from the patient, and it was also positive for *S. enterica*. Other microbiological tests were also performed, such as PCR screening of respiratory viruses and bacteria involved in atypical pneumonia (*Legionella pneumophila*, *Mycoplasma pneumoniae* and *Chlamydophila pneumoniae*), fungal culture from tracheal aspirate and urinary immunchromatography for *Streptococcus pneumoniae* and *L. pneumophila*. All these tests were negative. 

After the treatment with cefotaxime, the patient presented a favorable clinical course with all analytical parameters improving except for liver function abnormalities, and a new abdominal scan showed acute cholecystitis without further complications. Following the surgeons’ recommendations, conservative management was applied, and the patient clinically and analytically recovered. He received long-term antibiotics (19 days) with no other positive cultures obtained afterwards. Unfortunately, the neurological troubles aggravated, developing a non-convulsive status epilepticus. The patient, who remained in a vegetative state, was discharged from ICU after 30 days, and died 3 days later in the neurosurgery ward.

Four months before hospitalization in the ICU, the patient had been admitted to the nephrology ward of the same hospital with diagnosis of gastroenteritis and acute renal failure. At that time, *S. enterica* was already recovered from feces and blood cultures of the patient, after 24 h incubation in Hektoen agar (BioMérieux) or TSA agar, respectively. These isolates, unlike those obtained later, proved to be susceptible to ciprofloxacin, amoxicillin/clavulanic acid and piperacillin/tazobactan (Table 1). Accordingly, the patient was treated with intravenous ceftriaxone (1 gr/24 h for 6 days), and then with intravenous ciprofloxacin (200 mg/12 h for 3 days), before being discharged with complete recovery.

The original *S. enterica* isolate collected from feces of the patient was not available for further study. The remaining four isolates were subjected to whole-genome sequencing, performed with Illumina (Eurofins Genomics; Ebersberg, Germany). Sequence analysis was carried out using online tools (available at https://omictools.com/placnet-tool, accessed on 14 January 2021, and https://cge.cbs.dtu.dk/services, accessed on 12 March 2021), as previously reported [5]. The genomes, which consisted of a single chromosome without plasmids, were deposited in the NCBI GenBank database under accession numbers shown below.

By bioinformatic analysis, the isolates were assigned to *S. enterica* serovar 4,12:i:- (which is a monophasic variant of *S*. Typhimurium, lacking the second phase flagellin) [6,7], and to sequence type ST34. Disc diffusion assays, used to complement the microscan results, showed that the four sequenced isolates were resistant to ampicillin (10), streptomycin (10), sulfonamides (300) and tetracycline (30), with the amount in µg of each compound per disk (Oxoid) shown in parenthesis. This tetra-resistance pattern is characteristically associated with the ST34 monophasic variant of *S*. Typhimurium [6,7] and, as expected for this variant, the responsible genes were identified *in silico* as *bla*_TEM-1B_, *strA*, *strB*, *sul2* and *tet*(B) of chromosomal location [8,9]. However, as indicated before, the three isolates recovered in June 2020 from different samples of the patient were also resistant to ciprofloxacin, amoxicillin/clavulanic acid and piperacillin/tazobactam (Table 1). Previous therapy with ciprofloxacin and amoxicillin/clavulanic acid, the latter used for the treatment of a skin and soft tissue infection in primary care between the two hospital admissions, could have selected the additional resistances. A point mutation in the chromosomal gene *gyrB* (TCT to TTC), leading to a S464F amino acid substitution in the protein, was detected. The same mutation was previously reported in a clinical isolate of *S*. Typhimurium, which acquired decreased susceptibility to nalidixic acid, sparfloxacin and ciprofloxacin after therapy with the latter antimicrobial agent [10]. On the other hand, resistance to combinations of a β-lactam with a β-lactamase inhibitor has been associated with overproduction of a β-lactamase. In our isolates, the actual mechanism of TEM-1B overproduction: enhanced transcription of *bla*_TEM-1_ or increased copy number of the gene [11,12], is under investigation. Finally, in order to assess the genetic relationship between the isolates, the number of single nucleotide polymorphisms (SNP) was determined using the CSI phylogeny (https://cge.cbs.dtu.dk/services, accessed on 12 March 2021). The isolates were nearly identical, with the number of SNPs ranging from one to four, supporting the in vivo evolution of the new resistance properties.

## 3. Discussion

Respiratory tract infections caused by *S. enterica* are uncommon, although several cases have been reported mostly affecting immunocompromised patients, but also otherwise healthy hosts. The reported cases were mainly due to the invasive serovar Typhi, the two predominant NTS—*S*. Enteritidis, including its invasive non-typhoidal *Salmonella* (iNTS) variant endemic in Africa, and *S*. Typhimurium [13,14,15,16,17,18]. Other serovars have also been involved [13] but, to the best of our knowledge, this is the first report of a pneumonia caused by the emergent ST34 monophasic variant of *S. enterica* serovar Typhimurium. Isolates of this variant have increasingly been associated with human salmonellosis worldwide [7]. In the European Union, in 2019, the monophasic variant of *S*. Typhimurium ranked as the third most commonly reported serovar, only outnumbered by *S*. Enteritidis and biphasic *S*. Typhimurium [19].

As indicated before, third-generation cephalosporins and fluoroquinolones are drugs of choice for the management of invasive *Salmonella* infections in adults [3]. Thus, treatment-associated selection of fluoroquinolone resistance is a cause of concern. After the first bacteremia suffered by the patient during hospitalization at the nephrology ward, *S. enterica* could have reached the gallbladder through the descending route. Once there, bacteria could have survived the applied treatment, to later invade the intestine, blood stream and respiratory tract of the patient during the second admission of the patient to the hospital. The acquired ciprofloxacin resistance could have been crucial to bacterial survival within the gallbladder.

In contrast to fluoroquinolones, piperacillin/tazobactam is not recommended for the treatment of *Salmonella* infections. However, it is a drug combination used for the empirical management of patients with severe infections admitted to ICU, including nosocomial pneumonia. Therefore, infections caused by piperacillin/tazobactam-resistant bacteria are particularly relevant, because the pathogen would be able to evade the empirically administered therapy. It is of note that, although most monophasic ST34 isolates share the tetra-resistance phenotype characteristically associated with this clone (see above), resistances to critically important antibiotics, like third-generation cephalosporins, fluoroquinolones and colistin, have already been detected [5,7,20,21,22]. However, to the best of our knowledge, this is the first report of piperacillin/tazobactam resistance in isolates of the clone.

To conclude, the reported uncommon case of nosocomial pneumonia caused by *S. enterica* represents a paradigm of selective pressure leading to in vivo treatment-associated development of bacterial resistances, which, apart from facilitating the carrier state, can negatively affect the outcome of seriously ill patients in the ICU.

## Figures and Tables

**Table 1 antibiotics-11-00303-t001:** Clinical and microbiological characteristics of monophasic (4,12:i:-) ST34 isolates of *Salmonella enterica* recovered in a Spanish hospital from a patient with pneumonia.

Isolate ^a^	Origin	Resistance Phenotype ^b^ Acquired Resistance Genes *gyrB* Mutation (GyrB Substitution)	PTZ/NAL/CIP MIC ^b^ (µg/mL)	Pefloxacin ^c^ (mm)
HUCA_02174600 ^d^	Blood	AMP-STR-SUL-TET *bla*_TEM-1B_, *strA, strB*, *sul2*, *tet*(B)	≤8/8/0.047	Negative (24)
HUCA_020475219 ^e^	Tracheal Aspirate	AMP-AMC-PTZ-STR-SUL-TET-CIP *bla*_TEM-1B_, *strA*, *strB*, *sul2*, *tet*(B) TCT to TTC (S464F)	˃16/24/0.19	Positive (12)
HUCA_020475222 ^e^	Blood	AMP-AMC-PTZ-STR-SUL-TET-CIP *bla*_TEM-1B_, *strA, strB*, *sul2*, *tet*(B) TCT to TTC (S464F)	˃16/24/0.25	Positive (14)
HUCA_248014172 ^e^	Feces	AMP-AMC-PTZ-STR-SUL-TET-CIP *bla*_TEM-1B_, *strA, strB*, *sul2*, *tet*(B) TCT to TTC (S464F)	˃16/32/0.25	Positive (14)

^a^ HUCA, Hospital Universitario Central de Asturias. ^b^ AMP: ampicillin, AMC: amoxicillin/clavulanic acid, PTZ: piperacillin-tazobactam, STR: streptomycin, SUL: sulfonamides, TET: tetracycline, NAL: nalidixic acid, CIP: ciprofloxacin. ^c^ The pefloxacin 5 μg disk diffusion assay, recommended by EUCAST to screen for clinical fluoroquinolone resistance in *Salmonella*, was performed. ^d^ Isolate recovered during the first admission of the patient to the hospital. An isolate with the same resistance phenotype was recovered from feces at the same time, but it was not preserved for further analysis. ^e^ Isolates recovered during the second admission of the patient to the hospital.

## Data Availability

The genome sequences of the isolates were deposited in the NCBI GenBank database under accession numbers JAHMHX000000000 (HUCA_02174600), JAHMHY000000000 (HUCA_020475219), JAHMHZ000000000 (HUCA_020475222) and JAHMIA000000000 (HUCA_248014172).

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
