# Peer review of "Nosocomial Pneumonia Caused in an Immunocompetent Patient by the Emergent Monophasic ST34 Variant of Salmonella enterica Serovar Typhimurium: Treatment-Associated Selection of Fluoroquinolone and Piperacillin/Tazobactam Resistance"

_antibiotics, 2022, doi:10.3390/antibiotics11030303_

Round 1

Reviewer 1 Report

Dear academic editor: 
Vazquez et al. drafted a case report with the S.enterica with monophasic ST34 variant related pneumonia and septic shock. The authors provided the clinical course for the patient thoroughly. However, several points should be addressed
1. Please provide the evidence of immunocompetent of the subject because there is few result relevant to the patient. 
2.  Please give more clinical relevance on the variant and possible mechanisms of the S4 variants. For instance, did the patient receive unnecessary antibiotics for S. enterica in the first admission?
3. Could authors provide the literature reviews on S43 S. enteritis infection beyond the airway infection and their clinical outcomes? Because the patient's infection was not controlled.

Author Response

Dear Reviewer 1,

Please find enclosed our point by point answers to your comments and requirements, shown below each of them, preceded by an asterisk. We would like to thank you for your time and contribution to improving the manuscript.

Reviewer 1

Dear academic editor:
Vazquez et al. drafted a case report with the S.enterica with monophasic ST34 variant related pneumonia and septic shock. The authors provided the clinical course for the patient thoroughly. However, several points should be addressed
1. Please provide the evidence of immunocompetent of the subject because there is few result relevant to the patient.

* The patient was considered immunocompetent since he did not receive immunosuppressive therapies or suffered from any disease that compromised his immune system. This has been remarked in the new version of the manuscript.

  1. Please give more clinical relevance on the variant and possible mechanisms of the S4 variants. For instance, did the patient receive unnecessary antibiotics for S. enterica in the first admission?

* Do you mean ST34? For clinical relevance of this variant, please see references 7 and 19, and lines 160-163.

* The treatment during the first admission of the patient was correctly applied according to therapeutic guidelines. Considering that the patient had a bacteriemic infection, he was treated during ten days, first with an intravenous third generation cephalosporin and then with intravenous ciprofloxacin. Both drugs are recognized as standard choices for the treatment of Salmonella invasive infections.

  1. Could authors provide the literature reviews on S43 S. enteritis infection beyond the airway infection and their clinical outcomes? Because the patient's infection was not controlled.

* Sorry, we do not understand the meaning of the question. The manuscript deals with the ST34 monophasic variant of S. Typhimurium, not with S43 S. Enteritidis. Beyond the airway infection, the clinical outcomes of the ST34 variant were gastroenteritidis and bacteremia, according, like most other non-typhoid serovars of S. enterica (lines 41-45). Please also note that the infection could be controlled, and that the patient died due to aggravation of his neurological conditions (lines 93-98).

Reviewer 2 Report

The Authors describe a case of nosocomial pneumonia by S. enterica serovar typhimurium with specific gene sequencing and susceptibility analysis.

The case is interesting and quite well described. Some improvements can be added. In particular I suggest:

  • please better specify the diagnosis of pneumonia: ie add a line about the eventually inflammatory nature of the tracheal aspirate
  • please clarify the previous treatment (during the hospitalization at the Nefrology ward) in term of duration and dosage. FQ can favou the carrier state of salmonella spp in the gallbladder. Add a comment about this question
  • please add a comment in the discussion about the European a local susceptoblity pattern of salmonella
  • please make a English language and spell check (ie. correct "traqueal", abstract line 23)

Author Response

Dear Reviewer 2,

Please find enclosed our point by point answers to your comments and requirements, shown below each of them, preceded by an asterisk. We would like to thank you for your time and contribution to improving the manuscript.

Reviewer 2

The Authors describe a case of nosocomial pneumonia by S. enterica serovar typhimurium with specific gene sequencing and susceptibility analysis.

The case is interesting and quite well described. Some improvements can be added. In particular I suggest:

  • please better specify the diagnosis of pneumonia: ie add a line about the eventually inflammatory nature of the tracheal aspirate

* The patient presented respiratory failure and fever with leukocytosis, together with high procalcitonin levels (46 ng/dl; normal range 0.0-0.5 ng/dl). Pneumonia was confirmed by a chest radiography in which new infiltrates appeared in the right inferior lung lobe. Regarding microbiological diagnosis, Gram stain of the tracheal aspirate showed a single type of Gram negative bacilli, later identified as S. enterica, as well as a large number of leukocytes and a small number of epithelial cells. Other microbiological tests were performed such as PCR amplification with primers specific for detection of agents of atypical pneumonia (Legionella pneumophila, Mycoplasma pneumoniae and Chlamydophila pneumoniae) and respiratory viruses; fungal culture from tracheal aspirate; and urinary immunchromatography for Streptococcus pneumoniae and Legionella pneumophila. All these tests were negative. This information has been added to the new version of the manuscript (lines 60-63 and 86-90).

  • please clarify the previous treatment (during the hospitalization at the Nefrology ward) in term of duration and dosage. FQ can favou the carrier state of salmonella spp in the gallbladder. Add a comment about this question

* The patient was treated with intravenous ceftriaxone (1 gr/24h 6 days) iv), and then with ciprofloxacin (200 mg/12h 3 days iv). The dosages and duration of therapy were added to the new version of the manuscript (lines 104-107).

* FQ are used to eliminate the carrier state in typhoid serovars like S. Typhi, so it is unlikely to favour persistence of S. enterica in the gallbladder. In our opinion, after the first blood-stream infection during hospitalization of the patient in the nephrology ward, S. enterica could have reached the gallbladder via the descending route. Once there, it has apparently survived the applied treatment, to later originate the second intestinal infection, the second bacteremia and the respiratory tract infection. The acquired resistance to ciprofloxacin could have been crucial to survival. This information has been indicated in Discussion of the revised version (lines 166-172).

  • please add a comment in the discussion about the European a local susceptoblity pattern of salmonella

* A comment about the general susceptibility patterns of the European clone was added to the discussion, including relevant references. The local susceptibility patterns in our region coincide with those observed in any other region (lines 178-181; references 5, 7. 20-22).

  • please make a English language and spell check (ie. correct "traqueal", abstract line 23)

* Done as required.

Reviewer 3 Report

This article is a well-written case report of pneumoniae caused by Salmonella enterica. The manuscript could be improved by providing some additional details and addressing the points suggested below.

Major comments:

  1. It would be nice if the identification process for respiratory specimens (tracheal aspirate) could be written in detail. According to the “Clinical microbiology procedures handbook” reporting guideline, enteric GNB is identified only in the case of a single morphotype, but has it grown like that? Also, I would like to write in detail the agar (such as Salmonella-shigella agar) used for identification for each specimen (Blood, tracheal aspirate, stool) and how many colonies grown on solid agar for tracheal aspirate).
  2. Have all the causative pathogens of nosocomial pneumonia been excluded through appropriate laboratory testing? A detailed description is required. (e.g., C. pneumoniae (IgM/IgG), L. pneumophilia, Respiratory viruses, S. pneumoniae, M. pneumoniae etc.) Also, is it possible that tracheal aspirate sample was a blood-tinged sample?
  1. Please describe how long the patient has been on antibiotic treatment, and description of the follow up culture results (Line 77 and Lines 91-92).
  2. It would be better to describe the MIC change of the antibiotics related to the detected acquired resistance genes of the four isolates in the table or in the manuscript.

Minor comments:

  • Line 52: Please add the basic information about the patient (for example, age, sex, underlying disease etc.)
  • Line 59: Please insert the normal range of procalcitonin
  • Line 65: Please describe which parameter worsen and the blood biochemical test in detail.
  • Line 86: UCI, Isn't that a typo error?
  • Line 89 quinolone/fluoroquinolones means ciprofloxacin? It would be better to write clearly as ciprofloxacin.
  • Table 1: "NAL" Define the abbreviation completely.

                       Change “The strAB” to “strA, strB”.

Author Response

Dear Reviewer 3,

Please find enclosed our point by point answers to your comments and requirements, shown below each of them, preceded by an asterisk. We would like to thank you for your time and contribution to improving the manuscript.

Reviewer 3

This article is a well-written case report of pneumoniae caused by Salmonella enterica. The manuscript could be improved by providing some additional details and addressing the points suggested below.

Major comments:

  1. It would be nice if the identification process for respiratory specimens (tracheal aspirate) could be written in detail. According to the “Clinical microbiology procedures handbook” reporting guideline, enteric GNB is identified only in the case of a single morphotype, but has it grown like that? Also, I would like to write in detail the agar (such as Salmonella-shigella agar) used for identification for each specimen (Blood, tracheal aspirate, stool) and how many colonies grown on solid agar for tracheal aspirate).

* 109 colony forming units (CFU)/mL of S. enterica grew directly from the tracheal aspirate and in pure culture on agar McConkey. Blood cultures were subcultured in agar TSA where S. enterica was recovered; stool specimens were cultured in Hektoen agar. In all cases the recovered bacteria were identified by MALDI-TOF MS. This information has been added to the new version of the manuscript (lines 73-78).

  1. Have all the causative pathogens of nosocomial pneumonia been excluded through appropriate laboratory testing? A detailed description is required. (e.g., C. pneumoniae (IgM/IgG), L. pneumophilia, Respiratory viruses, S. pneumoniae, M. pneumoniae etc.) Also, is it possible that tracheal aspirate sample was a blood-tinged sample?

* Additional microbiological tests were performed, including PCR amplification with primers specific for detection of agents of atypical pneumonia (Legionella pneumophila, Mycoplasma pneumoniae and Chlamydophila pneumoniae) and respiratory viruses; fungal culture from the tracheal aspirate; and urinary immunchromatography for Streptococcus pneumoniae and Legionella pneumophila. All these tests were negative. This information has been added to the revised version of the manuscript (lines 86-90).

* The tracheal aspirate sample did not contain red blood cells, only leukocytes and a small number of epithelial cells. This information has also been included in the new version of the manuscript (lines 66-69).

  1. Please describe how long the patient has been on antibiotic treatment, and description of the follow up culture results (Line 77 and Lines 91-92).

* This information has been added to the manuscript. The patient was treated with ceftriaxone 1 gr/24h 6 days iv, and then with ciprofloxacin 200 mg/12h 3 days iv (lines 104-107).

  1. It would be better to describe the MIC change of the antibiotics related to the detected acquired resistance genes of the four isolates in the table or in the manuscript.

* The MICs of PTZ were added to the table, and the MICs to NAL and CIP are also shown.

Minor comments:

Line 52: Please add the basic information about the patient (for example, age, sex, underlying disease etc.).

* This information has been added, as requested.

  • Line 59: Please insert the normal range of procalcitonin

* Done as requested.

  • Line 65: Please describe which parameter worsen and the blood biochemical test in detail.

* Done as requested.

  • Line 86: UCI, Isn't that a typo error?

*This has been corrected.

  • Line 89 quinolone/fluoroquinolones means ciprofloxacin? It would be better to write clearly as ciprofloxacin.

* Done as requested.

  • Table 1: "NAL" Define the abbreviation completely.

* Done as requested.

  • Change “The strAB” to “strA, strB”.

* Done as requested.

Round 2

Reviewer 1 Report

Thank you for your the rapid reply on my question. The authors replied on the question 1 and 2 promptly and properly.  
About the question 3, my viewpoint is that the patient died in the ST34 related sepsis, since the ST34 was detected in the blood, feces and airway. The references provided by authors only provide the information about the epidemiology report about ST34. Is ST34 variance associated with prolonged ICU stay, higher risk of bloodstream infection or poor clinical outcome? ST34 was noted to be related to colistin - resistance strain and copper resistance. Such importance was not mentioned and therefore might not arouse the alert of the clinicians to may more attention on this stain. 

Author Response

Dear Reviewer 1,

Please find enclosed our point by point answers to your comments and requirements, shown below each of them, preceded by an asterisk. We would like to thank you again for your time and help in improving the manuscript.

Reviewer 1

Comments and Suggestions for Authors

Thank you for your the rapid reply on my question. The authors replied on the question 1 and 2 promptly and properly.
About the question 3, my viewpoint is that the patient died in the ST34 related sepsis, since the ST34 was detected in the blood, feces and airway. The references provided by authors only provide the information about the epidemiology report about ST34. Is ST34 variance associated with prolonged ICU stay, higher risk of bloodstream infection or poor clinical outcome? ST34 was noted to be related to colistin - resistance strain and copper resistance. Such importance was not mentioned and therefore might not arouse the alert of the clinicians to may more attention on this stain.

* The patient was admitted to ICU due to seizures and developed first nosocomial pneumonia associated with bacteremia and later on an acute cholecystitis. He received long term antibiotics and recovered completely from the infectious problems.

* The long ICU stay was due to the non-convulsive status epilepticus and the subsequent vegetative state, which finally led to his death in the neurology ward (lines 94-98/Revision 1-pdf).

* It is well known that nosocomial infections are associated with longer ICUs stays and higher rates of mortality, but nosocomial pneumonia by Salmonella enterica is extremely rare, and there are no previous data supporting that the ST34 clone has higher risk to cause bacteremia or respiratory infections.

* Please note that the ability of the ST34 clone to acquire resistance to critically important antibiotics in human medicine, including broad spectrum cephalosporins and colistin was already highlighted in the revised version (lines 178-181/Revision 1-pdf).

Reviewer 3 Report

Thank you for the opportunity to review your work.  In my second review I found the authors did an excellent job incorporating the previous comments into the revision. 

Author Response

Reviewer 3

Comments and Suggestions for Authors

Thank you for the opportunity to review your work. In my second review I found the authors did an excellent job incorporating the previous comments into the revision.

Dear Reviewer 3

* Thanks for your words and for helping us to improve the manuscript with your constructive comments and suggestions.